# Approaches to Medical Emergencies on Commercial Flights

**DOI:** 10.3390/medicina60050683

**Published:** 2024-04-23

**Authors:** Gopi Battineni, Antonio Arcese, Nalini Chintalapudi, Marzio Di Canio, Fabio Sibilio, Francesco Amenta

**Affiliations:** 1Clinical Research Centre, School of Medicinal and Health Products Sciences, University of Camerino, 62032 Camerino, Italy; gopi.battineni@unicam.it (G.B.); nalini.chintalapudi@unicam.it (N.C.);; 2Centre for Global Health Research, Saveetha Medical College, Saveetha Institute of Medical and Technical Sciences, Chennai 600077, India; 3CIRM SERVIZI S.r.l., 00144 Rome, Italy; aarcese@cirmservizi.it (A.A.); fsibilio@cirmservizi.it (F.S.); 4Research Department, International Radio Medical Centre (C.I.R.M.), 00144 Rome, Italy

**Keywords:** medical emergencies, travel medicine, commercial flights, first aid, passenger safety

## Abstract

In-flight medical incidents are becoming increasingly critical as passengers with diverse health profiles increase in the skies. In this paper, we reviewed how airlines, aviation authorities, and healthcare professionals respond to such emergencies. The analysis was focused on the strategies developed by the top ten airlines in the world by examining training in basic first aid, collaboration with ground-based medical support, and use of onboard medical equipment. Appropriate training of crew members, availability of adequate medical resources on board airplanes, and improved capabilities of dialogue between a flying plane and medical doctors on the ground will contribute to a positive outcome of the majority of medical issues on board airlines. In this respect, the adoption of advanced telemedicine solutions and the improvement of real-time teleconsultations between aircraft and ground-based professionals can represent the future of aviation medicine, offering more safety and peace of mind to passengers in case of medical problems during a flight.

## 1. Introduction

It has become a fact of life that air travel has become one of the most popular modes of transportation for people in today’s fast-paced world. Even though flying generally is a safe activity, there is always a risk of a medical emergency occurring on a commercial flight. On such occasions, the presence of well-trained and well-equipped medical personnel on board can significantly improve the chances of a positive outcome. The safety and well-being of passengers on commercial flights can be adversely affected by medical emergencies. Airlines and crews must implement effective measures when it comes to dealing with such incidents on their flights. There is basic health training provided to commercial flight crews, and in the event of an emergency, they have access to surgical kits on board [1]. As a result, they can assess the situation, assist, and request medical support from ground-based professionals using radio or satellite communication between the aircraft and the ground teams.

It may be possible for a doctor or another healthcare professional to offer assistance if they are present on board. To ensure that a passenger who has a medical condition receives the best possible care, aircraft crews, medical professionals, and ground support must communicate effectively at all times. Medical emergencies are handled in several ways on commercial flights, depending on the strategy and protocol that have been developed [2,3]. The crew members of some airlines are trained in how to deal with medical emergencies in case they arise [4]. The aim of these programs is to provide crew members with basic medical knowledge and skills, enabling them to assist passengers in need until professional medical assistance is available for them. As a result of this proactive approach, medical emergencies on commercial flights are handled more quickly with improved safety and well-being.

It has been established that different approaches have been developed for handling medical emergencies on commercial flights [5]. The ideal situation would be to have a dedicated medical professional on board who can handle medical emergencies in the event of an emergency [6]. Some airlines encourage medical professionals to identify themselves to the flight crew to assist with the situation. There is an advantage to this approach in that it enables the passenger to receive healthcare attention as soon as possible. Many airlines train their flight crews in the use of first aid only in the event of medical emergencies as part of their first-aid training [5,7,8]. Whenever a medical professional is not present, it is not guaranteed that the level of care given will be the same as when there is a medical professional present. Although flight attendants are trained in basic first aid, they are not able to provide medical assistance until the plane lands.

It can have a significant impact on the outcome of a medical emergency on a plane if trained medical professionals are on board [7,9]. When medical emergencies occur, airlines often encourage medical personnel present, such as doctors, nurses, or paramedics, to volunteer their services [10,11]. One study reported that a total of 121 society members presented 44.6% involvement in at least one in-flight medical emergency [12]. Cardiovascular (40.0%) and neurological disorders (17.8%) were the most common diagnoses, and emergency diversions occurred in 10.6% of cases. In addition to assessing the situation, administering first aid, and communicating with ground-based medical support, these individuals can provide invaluable assistance. Identifying and enlisting the assistance of any medical professionals aboard is imperative for the flight crew. The individuals should be provided with the necessary medical supplies and equipment to aid in their assessment and treatment of the affected individuals once they have been identified.

Using the onboard medical kit and working alongside flight attendants, trained medical professionals can efficiently stabilize a patient using the limited resources available on the aircraft. There is a need for an emergency medical kit to be carried during commercial flights as a precaution against medical emergencies. This type of kit usually consists of basic health supplies, such as bandages, pain relievers, and oxygen, according to the manufacturer [13]. However, these kits cannot take the place of professional medical care if a passenger in distress needs immediate relief. Medical professionals on board can help determine whether a diversion is needed or if some interventions should be prioritized in emergencies [8]. It is possible to receive the highest level of medical care on board a flight, but not all flights offer this level of care. In addition to providing immediate relief, a well-designed emergency medical kit is also able to offer immediate aid, but it cannot replace professional medical care.

By communicating with the flight crew, medical professionals can make more informed decisions and assist in managing the situation until landing. Additionally, they provide reassurance to both passengers and the affected individual, helping to maintain a calmer atmosphere during stressful times. The medical professional can continue to monitor and care for the patient until the aircraft lands or additional medical assistance is available. Furthermore, some airlines have agreements with facilities that provide ground-based medical consultations [14,15]. In the event of a medical emergency, these services can provide real-time assistance to the flight crews. Consequently, emergency medical situations can be dealt with in a variety of ways on commercial flights, with each having its own set of advantages and disadvantages.

The literature gap on medical concerns during flight journeys emphasizes the imperative for further research and collaboration among healthcare experts, the aviation industry, and policymakers. In this paper, we have reviewed the on-board medical services provided by the top 10 major global airlines (based on Wikipedia). We advocate for promoting informed decision-making and enhancing the safety and well-being of travelers. To ensure passengers’ health during flight journeys, we believe it is crucial to invest in research, clinical practice, and health education.

## 2. Methods

### 2.1. Data Collection 

The data were presented through a comprehensive review of the official websites and documentation provided by the top 10 selected airlines: Delta Air Lines, American Airlines, United Airlines, Lufthansa, Air France–KLM, Southwest Airlines, British Airways (part of International Airlines Group), Turkish Airlines, China Southern Airlines, and Air Canada. Criteria used in selecting the company included revenue, asset value, fleet size, and market capitalization.

### 2.2. Classification of Medical Services

The emergency and non-emergency medical services provided by each airline were categorized based on the information available on their respective websites. The classification included aspects such as the presence of a medical response team (MRT), availability of medical equipment (ME), provision of emergency training (ET), requirement of pre-flight medical evaluation (PME), allowance of medication onboard (MO), and provision of first aid (FA). The MRT staff is categorized both on the ground and in the air to ensure prompt and effective medical response during flights. Upon landing or during boarding, ground staff can offer immediate assistance, including airport medical personnel and paramedics. Additionally, airlines may provide medical equipment and kits on-board and have medical professionals among their crews. Basic first-aid training is typically available to flight attendants, which allows them to provide initial assistance. Medical volunteers can assist passengers in need, or the captain can contact medical professionals on the ground.

### 2.3. Analysis

Using the collected data, we analyzed the extent of medical services provided by each airline, both in emergency and non-emergency situations. During this analysis, patterns, trends, and discrepancies across the airline offerings were identified as well as medical assistance and resources available onboard. Data were compiled into a spreadsheet in a structured format, with rows representing airlines and columns representing medical services (MRT, ME, ET, PME, MO, FA). Descriptive statistics summarized data obtained. Each medical service was calculated based on frequencies and percentages across airlines. A cross-tabulation has been performed to identify trends, patterns, and discrepancies among airlines’ offerings. Using contingency tables, one examines the relationship between variables (e.g., airline and availability of medical care).

## 3. Results

Table 1 provides a summary of the medical responses available on board for various airlines categorized based on emergency and non-emergency situations. It offers the corresponding website links for each airline listed, facilitating access to further information and services provided by the respective airlines. The availability is denoted by a checkmark (✔) if available and a cross (✖) if not available.

The criteria for determining whether an airline provides emergency or non-emergency medical services likely include the availability of specific resources and personnel on board. For emergency medical services, criteria may involve the presence of an MRT, ME such as defibrillators or first aid kits, and emergency treatment capabilities. Having PME and oxygen supplies on hand is also considered essential in case of an emergency. A non-emergency medical service may include the presence of flight attendants trained in basic medical assistance and the availability of medical equipment and resources.

A team of doctors and nurses offers medical services on board, and they are experienced in dealing with health emergencies. Healthcare assistance can be obtained in both emergency and non-emergency situations that can be discussed further.

### 3.1. Emergency Medical Care

During flight, emergency medical services are designed to handle life-threatening situations. Medical response teams, equipment, and staff training are typically included in these services. 

➢
*Trained professionals*


Patients can benefit from highly trained individuals who can reduce uncertainty and delay in medical emergencies. Within minutes, they can provide life-saving care at the bedside. Aircraft that are air-conditioned but not pressurized to sea level can also increase the risk of medical events in flight. Air travel tends to be dry, so passengers with underlying health conditions, such as heart disease or respiratory issues, are at a greater risk of developing an in-flight medical condition [16]. Additionally, passengers with casts or plaster casts on their bodies, depending on whether they have just had surgery, are at the same risk. The unfamiliar environment, confined space, and prolonged immobilization can all lead to blood clots in a cabin environment. In companies like American Airlines, flight attendants are trained in basic first aid and can assist passengers in need, and some flights may also have healthcare professionals among the passengers who volunteer to help if there is a medical emergency.

According to Table 1, 8 of 10 airline companies report the presence of a medical response team, while Southwest and Air France—KLM do not report this information. Delta Airlines typically has trained crew members on board flights who can assist in case of medical emergencies [17]. They have also a program called the Volunteer Emergency Medical Service (VEMS), where medical professionals who are passengers on the flight can volunteer to assist if needed. American Airlines flight attendants are trained in basic first aid and can assist passengers in need, and some flights may also have healthcare professionals among the passengers who volunteer to help if there is a medical emergency [18]. Lufthansa, Austrian Airlines, and SWISS have published a book about in-flight medicine and aviation medicine as part of the “Doctor on Board” program [19]. Participants in the program will receive exclusive bag tags and will have the chance to attend a course run by Lufthansa’s Medical Service while acquiring Continuous Medical Education (CME) points. In the case of a medical emergency, flight attendants will be able to contact the doctor based on their medical specialization when a doctor registers with the program. A medical consultation involving several specialties is possible if several doctors are on board. Likewise, the on-board community of doctors in Air France—KLM offers a medical support service to ensure passengers’ comfort while traveling [20]. In the event of a medical emergency, the crew can quickly identify registered volunteer doctors. 

➢
*Equipment*


Medical kits are provided by all airlines with essential supplies and medications to address common health issues during flights. A first-aid kit typically contains basic first-aid supplies, pain-relief medications, and cardiac emergency equipment. It is essential to be familiar with the contents of medical kits and receive proper training [18]. Jochen Hinkelbein (2021) analyzed data from different European airlines to compare emergency medical equipment available and to show relevant differences in the available equipment [21]. A first aid kit (FAK; according to EU-OPS) was provided by all airlines. In particular, 18 out of 22 airlines (82%) reported having a doctor’s kit (DK) or an “Emergency Medical Kit” (EMK) on board. Overall, 86% of airlines (19/22) provided identical equipment in all aircraft of the fleet. The European Aviation Safety Agency (EASA) defines 36 different contents required in an EMK [22]. In total, 2 out of 22 airlines (9%) provided all the materials required for the EMK. Otherwise, four contents were not provided by the other airlines. There were missing advanced cardiac life support (ACLS) cards (71%), β-blockers (59%), rectal or oral sedatives (35%), and suction (35%) in most cases.

During a health emergency, the flight crew must be able to access medical kits quickly and provide them to professional healthcare personnel if on board. To ensure that the appropriate resources are available for emergency management, there must be clear communication about the contents of the medical kits and any specific requests from the medical professionals. It depends on the airline, the type of aircraft, and the regulations in each region whether this equipment is available on flights. In the event of a cardiac emergency, some airlines have automated external defibrillators (AEDs). In Table 2, we present information on airlines carrying AEDs on board and the status of various airlines. Passengers experiencing breathing difficulties can request supplemental oxygen from airlines. Possible emergency medications include pain relievers, antihistamines, and nausea medications. Healthcare professionals on the ground may be able to guide in medical emergencies via direct communication systems with airlines. In medical emergencies, flight attendants can assist passengers by performing CPR and basic first aid. The plane will land, and further emergency assistance can be obtained when the plane lands.

➢
*Emergency training*


The importance of addressing medical emergencies on board has been recognized by airlines. Crew members must be well trained and knowledgeable about medical emergencies. They must be familiar with the contents of the on-board first aid kit and be capable of assessing the situation quickly. It is reported that major airline companies have implemented comprehensive medical emergency programs that train their crew members to handle a variety of emergencies [4]. Basic life support, emergency breathing assistance, and the use of medical equipment are often covered in these programs [3,4,24,25]. All crew members undergo emergency training to ensure they are prepared for medical emergencies on board. The course covers first aid, cardiopulmonary resuscitation, and medical equipment. The situation is critical when a medical emergency occurs mid-flight. Crew members must be adequately trained and equipped with medical supplies and protocols in place for airline emergencies. In such a way, they can ensure that any medical emergency can be handled rapidly and effectively. Airline companies provide cabin crews with training so that they are capable of adequately assessing and responding to medical emergencies on flights [3,24]. As a result of this training, they will be equipped with the knowledge and skills they need to provide first aid, determine the severity of the condition, and communicate with medical professionals on the ground.

### 3.2. Non-Emergency Medical Services

Flight attendants provide non-emergency medical services to address situations that are not life-threatening. Some of these services may include:➢*Pre-flight medical evaluation*

For passengers who require medical treatment during their flight, a medical examination is required before departure. It is this method that will allow the identification and provision of appropriate medications during the flight in case a potential health issue arises [26]. Medical conditions like SARS-CoV-2 or recent surgeries may require clearance from a healthcare provider before flying, especially if the altitude or cabin pressure might affect them [27,28]. At Rome Fiumicino Airport, passengers were screened for SARS-CoV-2 infection, a strategy aimed at preventing the spread of the disease through COVID-free travel and revealed the airport’s role as a formidable sentinel station [27]. In the case of heart attacks, respiratory problems, or certain infectious diseases, a medical clearance may be required before flying. Pregnant passengers often have specific guidelines, especially in the latter stages of pregnancy. People in these cases are advised to consult their physician or healthcare provider before taking off.

➢
*Medical prescription*


A passenger with a prescription medication should be allowed to carry it on board with them. It is the policy of all airline companies to allow passengers to bring prescription medications on board in their carry-on luggage. It is, however, important to keep medications in their original containers during international travel to avoid security checks. According to China Southern Airlines, medication can be taken with certain restrictions, like a prescription label or a note from the physician to clarify its need [29]. Security personnel may ask about medication during screening, so passengers should be informed in advance [30]. Thorough research of a specific country’s guidelines or contacting its embassy or consulate before importing medications can prevent any complications. Especially for long flights, some medications may require specific storage conditions and consideration of factors like temperature sensitivity and access during flight. Individuals with complex medical needs or concerns should consult a healthcare provider before traveling to address any potential issues or adjustments.

➢
*Basic first aid*


Flight attendants are trained to provide basic first aid on board to ensure passengers’ safety and well-being. Cabin crew members can administer over-the-counter medications or bandages to wounds in case of minor injuries or illnesses during a flight. Flight attendants are equipped with basic first-aid kits to handle basic health issues including bandages, antiseptic wipes, adhesive tape, gauze pads, gloves, and other essential supplies for minor injuries [21]. For more serious cases or emergencies, some airlines have medical professionals on hand. It is the flight attendant’s responsibility to follow specific protocols if a medical emergency arises during the flight. Medical information and medications should also be carried on board by passengers with pre-existing conditions. 

## 4. Discussion

Medical emergencies on planes require fast communication and coordination between the crew and passengers, and the crew should keep passengers informed about the emergency regularly. During the initial stages of a medical emergency on a plane, the ability to effectively communicate and coordinate is essential for the well-being of the person affected. In this review paper, we have examined the importance of the role of trained personnel, the use of medical equipment, and the coordination with ground support in the context of a medical emergency on a plane by considering the top 10 global airlines.

When a medical emergency occurs on a plane, effective communication is the key to a successful outcome. For a timely and appropriate response to the situation, the flight crew, medical professionals, and the affected individual must communicate clearly and concisely. During a medical emergency, the flight crew plays an important role in initiating and maintaining communication. Medical professionals on board must be trained to recognize signs of distress and relay the information as soon as possible. Moreover, it is crucial to communicate clearly with the affected individual to obtain pertinent medical information, symptoms, and medications [31]. Medical professionals need this information to make informed treatment decisions. It is also important to remain calm and reassure the passengers that the situation is being handled competently by communicating effectively with them. Clear instructions and reports are essential for pilots, air traffic control, and cabin crew to navigate airspace, avoid collisions, and manage emergencies safely.

### 4.1. The Size of the Problem

Some published studies have provided more specific data on the numbers and frequency of medical emergencies during air travel [32,33,34]. In 2010 and 2011, data from the Lufthansa registry revealed common medical emergencies like diarrhea, nausea, vomiting, circulatory collapse, hypertension, stroke, and headache (including migraine) [33]. During flight, syncope, gastrointestinal events, respiratory, and neurological diagnostic groups were the most frequently encountered medical conditions or syndromes. In-flight medical emergencies were reported as 18.2 events per million passengers globally, based on 18 individual studies with around 1.5 billion passengers. The all-cause mortality rate was reported as 0.21 per million passengers [32]. It has been calculated that there were 11.1 diversions per 100,000 flights, and the average cost per unplanned emergency landing was between $15,000 and $893,000 [32].

Over 12 months, 131,890 international and domestic flight sectors transported more than 27 million passengers [34]. The total number of medical events per month was 296, resulting in 3555 incidents throughout the year. There was a 1:40 chance of a medical event on a flight, which was about 2.7%. Loss of consciousness (37%), or suspected cardiovascular events (12%), were the most common in-flight incidents classified as emergencies. A total of 6 out of the 915 emergencies resulted in fatalities. Medical incidents led to the diversion of twenty-one flights, which was less than 0.016% of all flights. In total, 52% of these diversions were linked to the suspicion of cardiac events.

### 4.2. Communication with Ground Staff

Flight attendants must communicate clearly to passengers during flight to ensure they understand safety instructions, emergency procedures, and general information [31,35,36]. As a result, passengers feel more secure and comfortable during their journey. Emergencies such as turbulence, medical incidents, or equipment malfunctions require clear and concise communication to pass instructions to passengers promptly and effectively. Communication with air traffic control is essential for pilots to receive instructions regarding flight paths, altitude changes, weather updates, and potential hazards. For coordinated tasks to be executed effectively and any issues to be addressed promptly, effective communication between the flight crew members is essential.

Medical professionals on the ground can offer via telecommunication systems guidance and support in managing the situation through protocols established by airlines. Upon landing, emergency medical services can be arranged to meet the flight, relay vital signs, and provide medical advice. Ground-based medical support, including emergency medical services (EMS) like pregnancy and medical professionals at control centers, complements the care provided on board [37]. When certain medical emergencies arise, they have access to more extensive medical resources, including specialized equipment, medications, and advanced care facilities. Providing ground-based medical support with vital information about the situation allows them to better prepare for the incoming patient and provide specific instructions or advice to manage the situation.

Medical support on the ground can help make critical decisions, such as if the flight should be diverted to the nearest airport for emergency medical care or if the situation can be managed until the flight reaches its destination. By relaying information from the flight crew, ground medical support can prepare for receiving medical facilities, alerting them to the incoming patient’s condition and ensuring an efficient response. Flight crews and medical support on the ground must maintain effective communication channels.

### 4.3. Utilization of Telecommunication Services

With telemedicine capabilities on board, pilots and ground-based medical professionals can consult directly and instantly during in-flight medical emergencies. A virtual bridge between the aircraft and medical experts on the ground can be created using technology that enables real-time audio, video, and data transmission. Flight crews can access medical support beyond their areas of expertise by incorporating telemedicine systems. In real time, they can share live video feeds, vital signs, and other relevant data with on-ground professionals. Using this exchange, medical experts on the ground can assess the situation visually, guide diagnoses, and provide precise treatment recommendations.

By using telemedicine, ground-based professionals have the opportunity to observe a patient’s condition, assess symptoms, and provide accurate advice on necessary steps. This will result in improved initial care for the affected passenger, thanks to the prompt implementation of these recommendations by the flight crew. In an emergency, telemedicine protocols and technologies should be standardized across airlines to ensure uniformity and compatibility. The ability to effectively integrate these telemedicine systems into in-flight emergency response procedures will also require training flight crews. It is ultimately hoped that integrating telemedicine capabilities on-board aircraft will enable passengers experiencing medical issues on board to have access to expert medical guidance on the ground, thereby potentially improving the quality of care and outcomes.

### 4.4. Data-Driven Approaches

Meanwhile, there is a need for a more complete analysis of data regarding in-flight medical emergencies to better prepare and allocate resources accordingly. Centralizing information about in-flight medical incidents, including the nature of the emergency, passenger demographics, types of interventions needed, and outcomes, would be highly beneficial. It is possible to identify prevalent medical conditions such as cardiovascular events, gastrointestinal issues, and neurological emergencies by analyzing these data.

The implementation of data-sharing capabilities could be beneficial in addition to voice and video communication. To help ground-based medical support make better decisions and provide more precise guidance, systems could allow passengers’ medical histories, vital signs, or readings from on-board medical equipment to be transferred. The standardization of communication protocols and procedures between airlines and aviation authorities would also make coordination more efficient. It could include clear guidelines regarding when to contact ground-based medical support, what information to convey, and how to integrate their guidance seamlessly into the in-flight response plan. This could result in significant improvement in flight crews’ ability to seek immediate, accurate, and comprehensive guidance from ground-based medical support during in-flight medical emergencies. Medical emergencies during flight can be better managed with the advancement of communication technology.

Airlines can tailor their medical kits and resources to better deal with the most commonly encountered medical emergencies by knowing the frequencies and patterns of these incidents. By using this data-driven approach, on-board medical kits can be optimized to contain the most relevant supplies and medications for managing prevalent conditions. A predictive model or risk assessment could also be developed by analyzing data on in-flight medical emergencies. To mitigate the risks associated with certain conditions, airlines can identify risks associated with flight duration, passenger demographics, and common triggers.

### 4.5. Limitations and Potential Bias

Limitations of this study include potential discrepancies in the availability and accuracy of information provided by the airlines on their websites. The reliance on publicly available data may introduce biases or gaps in the analysis, as the scope of information provided by airlines may vary. Additionally, the study’s focus on information available at a specific point in time may not capture changes or updates in airline policies or services. The analysis primarily considered data provided by the airlines themselves, which may not fully reflect the actual implementation or effectiveness of medical services during in-flight emergencies. Direct communication with airlines or additional research methods could provide further insights into the nuances of in-flight medical assistance.

There is a chance of potential bias in data analysis of the medical services provided by airlines. Because of selection bias, data may not represent a random sample of airlines globally, resulting in skewed results. The completeness and accuracy of the data may be affected by reporting bias if certain airlines are more transparent about their medical services. Medical service provision can be influenced by cultural or regulatory differences, while public access to information may be affected by publication bias. Providing incomplete or inaccurate information to respondents may cause results to be distorted, and limiting data sources to certain languages may lead to language bias. In addition, a time bias may occur if the data are not current, failing to capture changes in medical service offerings over time. To ensure the validity and generalizability of the analysis, it is crucial to recognize and address these potential biases.

## 5. Our Recommendations

The response to medical emergencies during flights still has room for improvement and presents opportunities for advancements. In addition to the basic first-aid training that flight attendants receive, additional training or periodic refreshers on the management of specific medical situations might be beneficial. This could involve scenarios involving a variety of medical conditions, mental health crises, or advanced life-support techniques. Training flight crew in medical emergencies requires a multifaceted approach that aims to improve their knowledge, skills, and confidence. In this respect, the availability of appropriate extensive epidemiological data on pathologies occurring on flights will help in approaching in the best way possible medical emergencies on airlines.

Specialized training sessions or workshops tailored to in-flight medical emergencies can provide insight and practical guidance to simulate medical emergencies, allowing flight crews to practice their skills and decision-making abilities in a controlled environment. By using e-learning platforms and technology-based training modules, these simulations can prepare attendants for real-life situations. This approach should include modules on effective communication, cultural sensitivity, and handling diverse passengers, among other things. On all flights, there should be standardized, well-equipped medical kits, which contain essential medications, equipment, and instructions for managing a wide range of medical emergencies. Medical kits should be standardized in flights to ensure they contain essential supplies to manage a wide range of medical emergencies.

Standards could be established and enforced by aviation authorities and/or international organizations, detailing specific items, quantities, and expiration dates. Healthcare professionals, emergency physicians, and healthcare organizations involved in developing standardized medical kits can ensure that the kits contain the necessary medications, equipment, and supplies. Provision of standardized medical kits with a periodic review and update system to integrate advances in medical practice, new medications, or equipment will help manage emergencies during flight.

The cabin crew should have easy access to medical kits, which should be well organized with clearly labeled items and clear instructions that facilitate quick and efficient access during emergencies. Flight crew members should receive comprehensive training on the contents and use of medical kits. In various emergencies, training should cover how to identify items, administer medications, and use equipment effectively. Several international aviation regulations and guidelines, including those set forth by the International Air Transport Association (IATA) [38] and the International Civil Aviation Organization (ICAO) [39], support the claim that flight crew members should receive comprehensive training on how to use medical kits. As part of these regulations, various provisions are made for medical emergencies and the safety of passengers during flights. Medical equipment and training requirements for flight crew members are outlined in the IATA’s Medical Manual and ICAO Annex 6—Operation of Aircraft [40]. It is essential to train crew members in handling medical emergencies, including using medical kits, as outlined in both documents.

A flight crew member is trained to administer basic first aid and use medical equipment but is not expected to perform advanced medical procedures beyond his or her training. Common medical emergencies, such as allergic reactions, minor injuries, and cardiac events, are typically managed with on-board medical kits containing basic medical supplies and medications. A non-healthcare professional, such as a flight crew member, cannot administer medications intravenously (IV). Healthcare professionals, in contrast, possess specialized training and qualifications for advanced medical interventions. Regardless of the severity of the medical emergency, flight crew members are trained to assess it, provide life support, and coordinate with ground-based medical professionals for further assistance, including the possibility of diverting the flight to a nearby airport if there is a serious medical problem.

Medical emergencies in flight require efficient communication between the flight crew and ground-based medical support. To communicate with ground-based support, airlines typically use voice communications through aircraft radios or satellite-based communication links. Telemedicine solutions are one method of integrating robust, real-time communication, which has been enabled by advances in technology. Medical professionals on the ground and flight crew could communicate directly via video or audio, facilitating immediate consultations and guidance. A telemedicine platform could allow medical experts on the ground to better assess and prescribe a treatment plan by sharing vital signs, images, or video feeds in real time. Furthermore, dedicated communication channels specifically designed for in-flight medical emergencies could streamline the process. There can be medical emergencies anywhere and at any time, even while on a flight.

## 6. Conclusions

Effective communication and coordination are indispensable during the initial stages of a medical emergency on a plane. Communication and coordination between the flight crew, medical professionals, and ground support are essential to ensuring the best outcome for the affected individual. In this paper, we examined the importance of recognizing and putting into practice these factors to make airlines more prepared for responding to medical emergencies, ultimately safeguarding the health and well-being of their passengers. Having the ability to respond promptly and effectively to such situations is crucial for airlines. The airline industry can ensure passenger safety by training cabin crew members, providing essential medical equipment, and establishing communication channels with medical professionals. The overall approach to handling medical emergencies on commercial flights includes collaboration with passengers and preserving legal obligations. To ensure passenger safety and crew well-being, medical emergencies on commercial flights must be handled promptly and effectively. To mitigate risks created by such incidents, airlines can implement effective communication strategies, train crew members on emergency response, and follow evacuation procedures. In-flight medical services based on telemedicine must be available for the safety and well-being of passengers during air travel. All passengers can rely on the flight crew and trained medical professionals for assistance in case of medical emergencies or minor ailments.

## Figures and Tables

**Table 1 medicina-60-00683-t001:** Summary of medical responses on board (✔: available and ✖: not available).

Airline	Flag	Emergency			Non-Emergency			Website
		MRT	ME	ET	PME	MO	FA	
Delta Air Lines	USA	✔	✔	✖	✖	✔	✔	https://www.delta.com/ (accessed on 29 November 2023)
American Airlines	USA	✔	✔	✖	✔	✔	✔	https://www.aa.com/homePage.do (accessed on 29 November 2023)
United Airlines	USA	✔	✔	✔	✔	✔	✔	https://www.united.com/en/us (accessed on 29 November 2023)
Lufthansa	Germany	✔	✔	✔	✖	✔	✔	https://www.lufthansa.com/it/en/homepage (accessed on 29 November 2023)
Air France–KLM	France	✖	✔	✖	✖	✔	✔	https://www.airfranceklm.com/en (accessed on 30 November 2023)
Southwest Airlines	USA	✖	✔	✖	✖	✔	✔	https://www.southwest.com/ (accessed on 30 November 2023)
British Airways (IAG)	UK	✔	✔	✔	✖	✔	✔	https://www.britishairways.com/travel/home/public/it_it/ (accessed on 2 December 2023)
Turkish Airlines	Turkey	✔	✔	✔	✔	✔	✔	https://www.turkishairlines.com/ (accessed on 2 December 2023)
China Southern	China	✔	✔	✖	✖	✖	✔	https://www.csair.com/ (accessed on 2 December 2023)
Air Canada	Canada	✔	✔	✔	✖	✔	✔	https://www.aircanada.com/it/en/aco/home.html (accessed on 2 December 2023)

MRT: medical response team, ME: medical equipment, ET: emergency training, PME: pre-flight medical evaluation, MO: medication onboard, FA: first aid.

**Table 2 medicina-60-00683-t002:** Airlines that carry AEDs (source: [23]). (✔: available and ✖: not available).

Airline	AED Defibrillator on Board
Air France	✔
Air Lingus	✔
Air New Zealand	✔
Aegean	✖
All Nippon Airlines	✔
British Airways	✔
Cathay Pacific	✔
EasyJet	✔
Emirates	✔
Etihad	✔
Finnair	✔
Japan Airlines	✔
Jet2	✔
KLM	✔
Lufthansa	✔
Norwegian	✔

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
