# Peer review of "Approaches to Medical Emergencies on Commercial Flights"

_medicina, 2024, doi:10.3390/medicina60050683_

Round 1

Reviewer 1 Report

Comments and Suggestions for Authors

Review of the Manuscript "Approaches to medical emergencies on commercial flights"

The objective of the study was to review how airlines, aviation authorities, and healthcare professionals respond to such emergencies. The manuscript is well-written, with clear and concise language that enhances the accessibility of the content. The authors' ability to communicate complex ideas in a straightforward manner contributes significantly to the manuscript's readability and overall impact.

Having thoroughly examined the content, I would like to pose a few clarifying questions to better understand certain aspects of the study.

·         All tables must be self-explanatory, i.e. every abbreviation should be explained in the footnotes of the tables.

·         How did you analyze the obtained data, by which statistical method?

·         Are there any potential biases that could be delineated in the analyzing process?

·         In Table 1: what were the criteria for determining if an airline provides emergency or non-emerg. med. service?

·         Do your recommendations align the current industry standards?

Kindly incorporate the responses within the manuscript to augment its overall quality.

Author Response

First of all, let us thank the reviewer for the constructive comments that helped us really to make more improvements to the current manuscript. Please find our response to each concern that was raised.

  1. All tables must be self-explanatory, i.e. every abbreviation should be explained in the footnotes of the tables.

The explanations for Table 1 and 2 were provided in the revised version. See the highlighted text P3 (L 131-135 & 207-208).

  1. How did you analyse the obtained data, and by which statistical method?

The procedure for data analysis is mentioned in the revised version. Please see Lines 120-129 of the revised manuscript.

  1. Are there any potential biases that could be delineated in the analysing process?

The limitations and potential biases were added in the revised version. Please refer to the section 4.4 of the revised manuscript.

  1. In Table 1: what were the criteria for determining if an airline provides emergency or non-emerg? med. service?

The criteria for selecting emergency and non-emergency were presented in the revised version. Please see the lines 136-142 of the revised manuscript.

  1. Do your recommendations align with the current industry standards?

Please see the section 5 of the revised manuscript.

We hope that all comments are addressed adequatly and looling forward to have final opinion on revised version 

Reviewer 2 Report

Comments and Suggestions for Authors

Respected! I congratulate you on your work. My comments are added in PDF file. Although my main concern is very lax approach to the study matter and lack of specific findings. Very vague.

Comments on the Quality of English Language

minor modifications needed

Author Response

First of all, let us thank him for the suggestive comments that helped us really to make more improvements to the current manuscript. Please find our reply to the concerns raised.

Respected! I congratulate you on your work. My comments are added in a PDF file. Although my main concern is very lax approach to the study matter and the lack of specific findings. Very vague.

With all due respect, we want to make sure that this work is to present our perspective on how airlines are responding to emergencies and nonemergencies. The criteria for airline selection and data compilation approaches were clearly described in the revised version. The responses to comments that were raised in the pdf file are outlined in yellow in the revised manuscript. 

We hope to have answered satisfactorily to the reviewer’s comments and that the revised manuscript which is herewith enclosed is now acceptable for publication.

I am looking forward to hearing from you and I would like to the thank you in advance for your attention and your time.